# Learning to Move with Affordance Maps

**William Qi**[1]    **Ravi Teja Mullapudi**[1]    **Saurabh Gupta**[2]    **Deva Ramanan**[1]
[1] Carnegie Mellon University    [2] UIUC

## Abstract

The ability to autonomously explore and navigate a physical space is a fundamental requirement for virtually any mobile autonomous agent, from household robotic vacuums to autonomous vehicles. Traditional SLAM-based approaches for exploration and navigation largely focus on leveraging scene geometry, but fail to model dynamic objects (such as other agents) or semantic constraints (such as wet floors or doorways). Learning-based RL agents are an attractive alternative because they can incorporate both semantic and geometric information, but are notoriously sample inefficient, difficult to generalize to novel settings, and are difficult to interpret. In this paper, we combine the best of both worlds with a modular approach that *learns* a spatial representation of a scene that is trained to be effective when coupled with traditional geometric planners. Specifically, we design an agent that learns to predict a spatial affordance map that elucidates what parts of a scene are navigable through active self-supervised experience gathering. In contrast to most simulation environments that assume a static world, we evaluate our approach in the VizDoom simulator, using large-scale randomly-generated maps containing a variety of dynamic actors and hazards. We show that learned affordance maps can be used to augment traditional approaches for both exploration and navigation, providing significant improvements in performance.

## 1 Introduction

The ability to explore and navigate within a physical space is a fundamental requirement for virtually any mobile autonomous agent, from household robotic vacuums to autonomous vehicles. Traditional approaches for navigation and exploration rely on simultaneous localization and mapping (SLAM) methods to recover scene geometry, producing an explicit geometric map as output. Such maps can be used in conjunction with classic geometric motion planners for exploration and navigation (such as those based on graph search).

However, geometric maps fail to capture dynamic objects within an environment, such as humans, vehicles, or even other autonomous agents. In fact, such dynamic obstacles are intentionally treated as outliers to be ignored when learning a geometric map. However, autonomous agents *must* follow a navigation policy that avoids collisions with dynamic obstacles to ensure safe operation. Moreover, real-world environments also offer a unique set of affordances and semantic constraints specific to each agent: a human-sized agent might fit through a particular door, but a car-sized agent may not; similarly, a bicycle lane may be geometrically free of obstacles, but access is restricted to most agents. Such semantic and behavioral constraints are challenging to encode with classic SLAM.

One promising alternative is *end-to-end* reinforcement learning (RL) of a policy for exploration and navigation. Such approaches have the potential to jointly learn an exploration/navigation planner together with an internal representation that captures both geometric, semantic, and dynamic constraints. However, such techniques suffer from well-known challenges common to RL such as high sample complexity (because reward signals tend to be sparse), difficulty in generalization to novel environments (due to overfitting), and lack of interpretability.

We advocate a hybrid approach that combines the best of both worlds. Rather than end-to-end learning of both a spatial representation and exploration policy, we apply learning only "as needed". Specifically, we employ off-the-shelf planners, but augment the classic geometric map with a *spatial affordance map* that encodes where the agent can safely move. Crucially, the affordance map is learned through self-supervised interaction with the environment. For example, our agent can

*discover* that spatial regions with wet-looking floors are non-navigable and that spatial regions that recently contained human-like visual signatures should be avoided with a large margin of safety. Evaluating on an exploration-based task, we demonstrate that affordance map-based approaches are far more sample-efficient, generalizable, and interpretable than current RL-based methods.

Even though we believe our problem formulation to be rather practical and common, evaluation is challenging in both the physical world and virtual simulators. It it notoriously difficult to evaluate real-world autonomous agents over a large and diverse set of environments. Moreover, many realistic simulators for navigation and exploration assume a static environment (Wu et al., 2018; Savva et al., 2017; Xia et al., 2018). We opt for first-person game-based simulators that populate virtual worlds with dynamic actors. Specifically, we evaluate exploration and navigation policies in VizDoom (Wydmuch et al., 2018), a popular platform for RL research. We demonstrate that affordance maps, when combined with classic planners, dramatically outperform traditional geometric methods by 60% and state-of-the-art RL approaches by 70% in the exploration task. Additionally, we demonstrate that by combining active learning and affordance maps with geometry, navigation performance improves by up to 55% in the presence of hazards. However, a significant gap still remains between human and autonomous performance, indicating the difficulty of these tasks even in the relatively simple setting of a simulated world.

## 2 Related Work

**Navigation in Classical Robotics.** Navigation has classically been framed as a geometry problem decomposed into two parts: mapping and path planning. Inputs from cameras and sensors such as LiDARs are used to estimate a geometric representation of the world through SLAM (or structure from motion) techniques (Thrun et al., 2005; Cadena et al., 2016). This geometric representation is used to derive a map of traversability, encoding the likelihood of collision with any of the inferred geometry. Such a map of traversability can be used with path planning algorithms (Kavraki et al., 1996; Canny, 1988; LaValle, 1998) to compute collision-free paths to desired goal locations. Navigation applications can be built upon these two primitives. For example, exploration of novel environments can be undertaken by sampling point goals in currently unknown space, planning paths to these point goals, and incrementally building a map using the sensor measurements along the way (also known as frontier-based exploration (Yamauchi, 1997)). Such an approach for exploration has proven to be highly effective, besting even recent RL-based techniques in static environments (Chen et al., 2019), while relying on classical planning (Fiorini & Shiller, 1998).

**Semantics and Learning for Navigation.** Taking a purely geometric approach to navigation is very effective when the underlying problem is indeed geometric, such as when the environment is static or when traversability is determined entirely by geometry. However, an entirely geometric treatment can be sub-optimal in situations where semantic information can provide additional cues for navigation (such as emergency exit signs). These considerations have motivated study on semantic SLAM (Kuipers & Byun, 1991), that seeks to associate semantics with maps (Bowman et al., 2017; McCormac et al., 2017), speed up map building through active search (Leung et al., 2008), or factor out dynamic objects (Bescos et al., 2018).

In a similar vein, a number of recent works also investigate the use of learning to solve navigation tasks in an end-to-end manner (Zhu et al., 2017; Gupta et al., 2017; Mirowski et al., 2017; Fang et al., 2019; Shrestha et al., 2019), built upon the theory that an agent can automatically learn about semantic regularities by directly interacting with the environment. Semantics have also been used as intermediate representations to transfer between simulation and the real world (Müller et al., 2018). While such use of learning is promising, experiments in past work have focused only on semantics associated with static maps. Instead, we investigate the role of semantics in dynamic environments, and in scenarios where the notion of affordance goes beyond simple geometric occupancy.

Another recent approach (Mirchev et al., 2018) introduces a method of learning generalized spatial representations for both exploration and navigation, employing an attention-based generative model to reconstruct geometric observations. Planning for navigation occurs in belief space, in contrast to the metric cost maps (incorporating both semantics and geometry) used in our work.

**Hybrid Navigation Policies.** While learning-based methods leverage semantic cues, training such policies can be sample inefficient. This has motivated the pursuit of hybrid policy architectures that

combine learning with geometric reasoning (Gupta et al., 2017; Bhatti et al., 2016) or known robot dynamic models (Bansal et al., 2019; Kaufmann et al., 2018; Müller et al., 2018). Our work also presents a hybrid approach, but investigates fusion of a learned mapper with analytic path planning.

**Self-Supervised Learning.** Recent work in robotics has sought to employ self-supervised learning (Pinto & Gupta, 2016) as an alternative to end-to-end reward-based learning. Hadsell et al. (2009) and Bruls et al. (2018) employ passive cross-modal self-supervision to learn navigability (from stereo to monocular images, and from LiDAR to monocular images, respectively). In contrast, we learn through active interaction with the environment. Thus, our work is most similar to that of Gandhi et al. (2017), though we learn *dense* traversability predictions for long range pathplanning, rather than short-range predictions for collision avoidance.

**Navigation in Dynamic Environments.** Finally, a number of other works develop specialized techniques for navigation in dynamic environments, by building explicit models for other agents' dynamics (Chen et al., 2018; Kretzschmar et al., 2016; Paden et al., 2016). In contrast, by generalizing our definition of traversability beyond geometry alone, we can automatically capture the dynamics of other agents implicitly and jointly with other environmental features.

# 3 APPROACH

Our goal is to build an agent that can efficiently explore and navigate within novel environments populated with other dynamic actors, while respecting the semantic constraints of the environment. The scenario we consider is a mobile agent capable of executing basic movement macro-actions. The agent is equipped with a RGBD camera and some form of proprioceptive feedback indicating wellbeing (e.g bump sensor, wheel slip, game damage). We assume that the agent is localized using noisy odometry and that depth sensing is also imperfect and noisy. At test time, the agent is initialized within a novel environment containing an unknown number of dynamic and environmental hazards. Furthermore, we assume that the exact dimensions of the agent and nature of affordances provided by entities within the environment are not initially known.

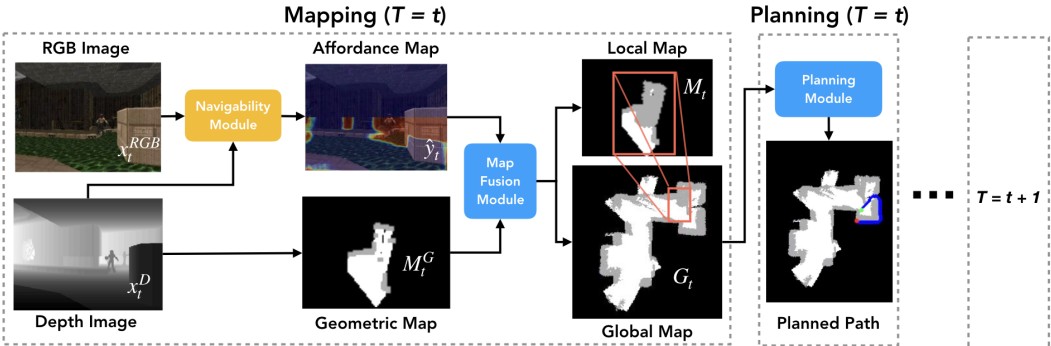

Figure 1: Overview of our proposed architecture for navigation. RGBD inputs $x_t$ are used to predict affordance maps $\hat{y}_t$ and transformed into egocentric navigability maps $M_t$ that incorporate both geometric and semantic information. In the example shown, $M_t$ is labelled as non-navigable in regions near the monster. A running estimate of the current position at each time step is maintained and used to update a global, allocentric map of navigability $G_t$ that enables safe and efficient planning.

We propose a modular approach to tackle this problem, adopting a classical pipeline of map building and path planning. Figure 1 shows an overview of this pipeline, which builds a navigability map using both geometric and semantic information, as opposed to traditional methods that rely on geometry alone. Our main contribution, shown in Figure 2, is a method for predicting which parts of a scene are navigable by actively leveraging the feedback sensor to generate partially-labeled training examples. We then use the labeled samples to train a model which predicts a per-pixel affordance map from the agent's viewpoint. At evaluation time, the outputs from the learned module are combined with geometric information from the depth sensor to build egocentric and allocentric representations that capture both semantic and geometric constraints. The fused representations can

then be used for exploration/navigation by employing traditional path planning techniques, enabling safe movement even within dynamic and hazardous environments.

## 3.1 NAVIGABILITY MODULE

Given a scene representation $x$ captured by a RGBD camera, our goal is to train a module $\pi$ that labels each pixel with a binary affordance value, describing whether the corresponding position is a valid space for the agent to occupy and forming a segmentation map of "navigability" $y$. We can encode this understanding of the environment by training an image segmentation model in a supervised fashion. However, training such a model requires a set of labeled training images $D = [(x_1, y_1), ...(x_n, y_n)]$ where each pixel is annotated for navigability. Traditionally, obtaining such a set of labels has required dense annotation by an oracle (Cordts et al., 2016), at a cost that scales linearly with the amount of data labeled. These properties have generally limited applications to domains captured by large segmentation datasets (Lin et al., 2014; Zhou et al., 2017) that have been curated using hundreds of hours of human annotation time. We address this problem by employing a self-supervised approach to generate partially labeled examples $\tilde{y}$, in place of oracle annotation.

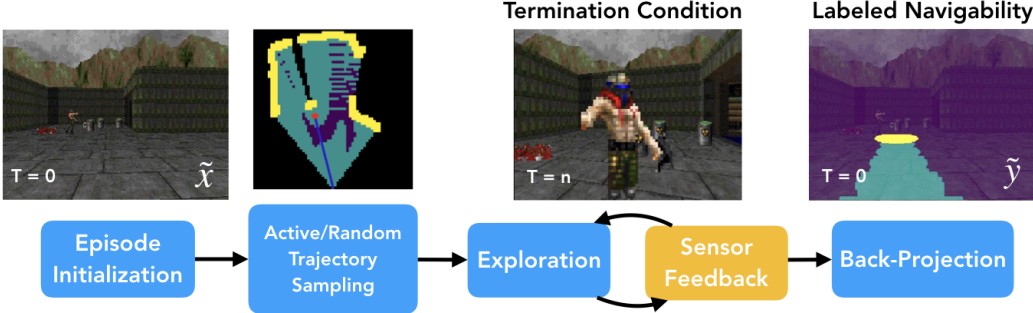

Figure 2: Overview of self-supervised labeling for navigability training pairs $(\tilde{x}, \tilde{y})$. The agent performs a series of walks along random or planned trajectories within the environment. Affordance information collected from each walk is back-projected onto pixel-level labels in the agent's POV from previous time steps. Sampling over a variety of maps allows for the collection of a visually and semantically diverse set of examples $\tilde{D}$ that can be used to train a navigability module $\pi$. This figure illustrates the generation of a negative example, with the agent contacting a dynamic hazard.

**Self-Supervision.** We generate labeled affordance data in a self-supervised manner through continuous interactive exploration by the agent; this algorithm makes use of RGBD observations $x$, readings from a feedback sensor $s$, and a history of actions $a_t$ executed over time. In each episode, the agent is initialized at a random location and orientation within a training environment. The agent selects a nearby point and attempts to navigate towards it. Labeled training data is generated based on whether or not the agent is able to reach this point: every location that the agent successfully traverses during its attempt is marked as navigable, while undesirable locations (e.g. bumping into obstacles, loss of traction, loss in health, getting stuck) are marked as non-navigable. These locations in world space are then back-projected into previous image frames using estimated camera intrinsics, in order to obtain partial segmentation labels (examples of which are visualized in Figure 3). Pixels for which there are no positive or negative labels are marked as unknown. A more detailed discussion about the real-world applicability of this approach can be found in Appendix A.4.

**Dense Labels.** Backprojection of affordance labels produces a dense set of *pixelwise* labels for observations at past time steps. Importantly, even without spatio-temporal inputs, this enables the training of models which incorporate safety margins to account for motion, as the future position of dynamic actors is encoded within labelled views from the past (discussed further in Appendix A.3). In contrast, most RL-based methods return only a single sparse scalar reward, which often leads to sample inefficient learning, potentially requiring millions of sampling episodes (Zhu et al., 2017). Furthermore, our generated labels $\tilde{y}$ are human interpretable, forming a mid-level representation that improves interpretability of actions undertaken by the agent.

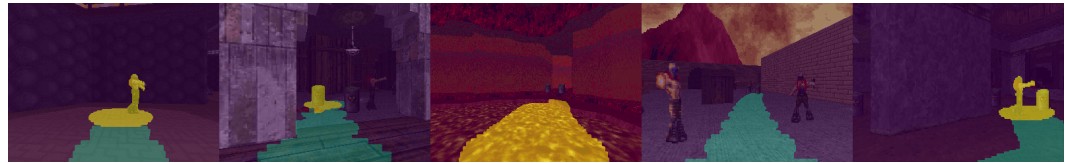

Figure 3: Examples of samples labeled through back-projection (navigable area labeled in green, non-navigable in yellow, and unknown in purple). The first three examples show negative examples, labeled by damage from monster, impediment of movement by barrel, and damage taken from environmental hazard respectively. The fourth illustrates successful traversal between monsters and the fifth shows an example collected along a minimum cost path as part of an active learning loop.

**Navigability Segmentation.** The collected samples $\tilde{D}$ are used to train a segmentation network such as UNet (Ronneberger et al., 2015), allowing for generalization of sampled knowledge to novel scenarios. A masked loss function $L_{mask} = K \odot L_{BCE}(\hat{y}, y)$ based on binary cross-entropy is employed to ensure that only labeled, non-unknown points $K$ within each example contribute to the loss. Given enough training data, the navigability module is capable of generating segmentation maps that closely approximate ground truth navigability, even in previously unseen environments.

**Active Trajectory Sampling.** In order to further improve sample efficiency, we can employ model uncertainty to actively plan paths during sampling episodes so as to maximize label entropy along traversed trajectories. Intuitively, many semantically interesting artifacts (such as environmental hazards) are rare, making it difficult to learn a visual signature. In these cases, sampling can be made more efficient by intentionally *seeking* out such artifacts. This can be achieved by first collecting a small number ($n$) of samples using random walks and training a seed segmentation model. Using the seed model, we then predict an affordance map $\hat{y}$ during the first step of each subsequent episode and use it to construct a cost map for planning, with values inversely proportional to the prediction uncertainty (defined as the entropy of the predicted softmax distribution over class labels) at each position. Planning and following a minimal-cost path in this space is equivalent to maximization of label entropy, as the agent will attempt to interact most with highly uncertain areas. Once an additional $n$ samples have been actively collected using this strategy, the model is retrained using a mixture of all samples collected so far, and the sample/train loop can be repeated again. We find that active learning *further* increases our sample efficiency, requiring fewer sampling episodes to learn visual signatures for hazards and dynamic actors (example shown in Figure 3 rightmost).

### 3.2 MAP CONSTRUCTION

While some hazards can only be identified using semantic information, geometry provides an effective and reliable means to identify navigability around large, static, obstacles such as walls. To capture both types of constraints, we augment our predicted semantic maps with additional geometric information when constructing the projected navigability cost maps $M$ and $G$ used for planning. As the agent moves around the environment, observed depth images are used to construct local, egocentric occupancy maps at each time step, incorporating only geometric information. By reading depth values from the center scanline of the depth image, projecting into the XY-plane, and marking the corresponding cells as non-navigable, a map of geometric obstacles $M_t^G$ can be obtained. As the exact dimensions and locomotion capabilities of the agent are unknown, only depth values returned by the center scanline are known to be obstacles with certainty.

**Map Fusion.** Given a pixel-wise affordance map $\hat{y}_t$ obtained from the navigability module and a local, egocentric geometric map $M_t^G$, the two inputs can be combined using a fusion module $F(\hat{y}_t, M_t^G)$ to form a single local navigation cost map $M_t$ that incorporates both semantic and geometric information. To do so, the segmentation map $\hat{y}_t$ is first projected into the 2D plane using estimated camera intrinsics, forming an egocentric navigability map $M_t^S$. Cells marked as obstacles by $M_t^G$ are also marked as impassable within $M_t$, with remaining cells in free space assigned cost values inversely proportional to the confidence of navigability provided by $\hat{y}_t$. Finally, $M_t$ is used to update a global, allocentric map $G_t$ of navigability at the end of each time step.

### 3.3 PLANNING

Given the global navigability map, path planning can be tackled using classical algorithms such as A*, as all required semantic and geometric information is encoded within the map itself. Additionally, since both $M_t$ and $G_t$ are updated at every time step, dynamic hazards are treated as any other obstacle and can be avoided successfully as long as paths are re-planned at sufficiently high frequency. Our work is agnostic to the choice of planning algorithm and our semantic maps can also be employed with more sophisticated planners, though for simplicity we evaluate using A*.

## 4 EXPERIMENTS

We perform our evaluation in simulation using VizDoom, as it allows for procedural generation of large, complex 3D maps that contain a variety of dynamic actors and semantic constraints in the form environmental hazards. Although prior work (Savinov et al., 2018) on navigation has also relied on VizDoom, evaluation has been restricted to a small set of hand designed maps without any dynamic actors or semantic constraints. We evaluate the effectiveness of incorporating learned affordance maps to tackle two difficult tasks: novel environment exploration and goal-directed navigation.

### 4.1 EXPERIMENTAL SETUP

We conduct our experiments within procedurally-generated VizDoom maps created by the Oblige (Apted, 2017) level generator, which enables construction of training and test maps containing unique, complex, and visually diverse environments. Each generated map is large, containing a variety of dynamic hazards (such as monsters) and environmental hazards (such as lava pools), in addition to static obstacles (such as barrels) and areas where a geometry-affordance mismatch exists (such as ledges lower than sensor height, but beyond the movement capabilities of the agent). We generate a collection of 60 training and 15 test maps and further categorize the 15 test maps as either hazard-dense or hazard-sparse, based on concentration of hazards within the initial exploration area.

**Observation and Action Space.** We assume that the agent's RGBD camera returns a regular RGB image with a $60°$ field of view and an approximately-correct depth image that records the 2D Euclidean distance of each pixel from the camera in the XY plane (due to the 2.5D nature of the Doom rendering engine). The feedback sensor returns a scalar value corresponding to the magnitude of damage received by the agent while executing the previous action (some hazards are more dangerous than others). The action space is limited to three motion primitives: move forward, turn left, and turn right; only one action can be executed at each time step. Localization is imperfect and achieved through odometry from noisy measurements, with approximately 2% error.

### 4.2 SAMPLE-EFFICIENT EXPLORATION USING AFFORDANCE MAPS

We quantitatively evaluate exploration performance by measuring the total amount of space observed within a particular environment over time, approximated by the total surface area of the constructed global map. Each episode of evaluation terminates after 2000 time steps or after receiving a total of 100 damage during exploration, whichever occurs first. Agents receive 4 damage per time step when coming into contact with dynamic hazards and 20 damage for environmental hazards.

**Frontier-Based Exploration.** As a classical, non-learning baseline, we compare against a variant of frontier-based exploration (Yamauchi, 1997; Dornhege & Kleiner, 2013). This approach relies purely on geometry, updating a global map $G_t$ at every step using the projected scanline observation $M_t^G$ from the current POV. A close-by goal from within the current "frontier region" is selected and a path towards it is re-planned (using A*) every 10 steps as the map is updated. Once the selected goal has been reached or the goal is determined to no longer be reachable, the process is repeated with a newly-selected goal. Although dynamic actors can be localized using geometry alone, they are treated as static obstacles in the cost map, relying on frequent re-planning for collision avoidance.

**RL-Based Exploration.** We also compare against a state-of-the-art deep RL-based approach (Chen et al., 2019) for exploration that is trained using PPO (Schulman et al., 2017) and incorporates both geometric and learned representations. We implement an augmented variant of the method proposed by (Chen et al., 2019), adding an additional depth map $x_t^D$ to the 3 original inputs: the current RGB

observation $x_t^{RGB}$, a small-scale egocentric crop of $G_t$, and a large-scale egocentric crop of $G_t$. We evaluate this approach using hyper-parameters identical to those proposed by the original authors, with the only exception being the addition of a new penalty in the reward that is scaled by the amount of damage received at each time step. We report the mean performance obtained by the best model from each of 3 training runs (2M samples each), with access to the full 60 map training set.

**Affordance-Augmented Frontier Exploration.** To evaluate the efficacy of our proposed representation, we augment the frontier-based approach with semantic navigability maps obtained from affordance predictions; all other components (including goal selection and path planning) are shared with the baseline. We collect approximately 100k total samples across the 60 training maps in a self-supervised manner and train the navigability module for 50 epochs using the collected dataset; a ResNet-18-based (He et al., 2016) UNet (Ronneberger et al., 2015) architecture is employed for segmentation. Episodic sample goals are selected randomly from within the initial visible area and simple path planning is employed, with the agent always taking a straight line directly towards the goal. Back-projection is performed using game damage as a feedback mechanism, with the size of negative labels corresponding to the magnitude of damage received. At test time, we use estimated camera intrinsics to project output from the navigability module into the 2D plane. Additional experimental details are discussed in Appendix A.1.

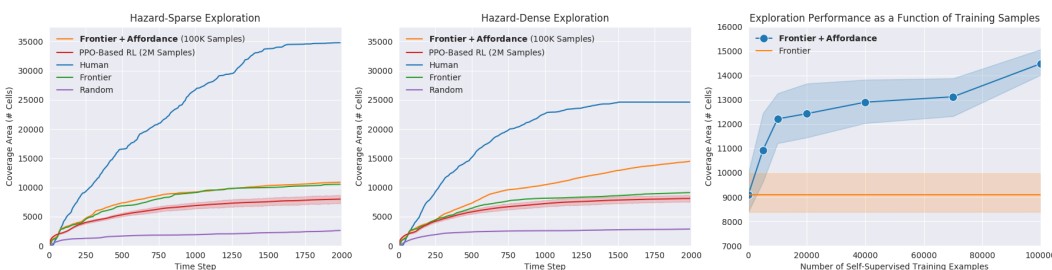

Figure 4: Comparison of exploration performance across all evaluated approaches in hazard-dense **(Left)** and hazard-sparse environments **(Center)**, plotted as a function of area observed over time. **(Right)** Comparison of final exploration coverage achieved by affordance-augmented frontier exploration, trained using varying amounts of self-supervised training data. All plots report mean performance measured over 5 test runs, shaded areas indicate the range of measured values, and RL results are reported as mean performance of the best model from each of 3 training runs.

Inside hazard-sparse environments (Figure 4 left), agents generally don't encounter hazards within the first 2000 time steps, placing increased emphasis on goal selection over hazard avoidance. In this setting, augmenting the frontier-based approach with affordance maps does not provide significant improvements, as in the absence of semantic hazards, the two methods are functionally equivalent. In line with previous work (Chen et al., 2019), the PPO-based RL approach also fails to beat the frontier baseline, likely due to the heavy emphasis placed on exploration policy. Without taking a high-level representation of the global map as input, it is difficult for a RL-based approach to plan over long time horizons, causing the agent to potentially re-visit areas it has already seen before. Finally, we note that humans are much better at both goal selection and hazard avoidance, managing to explore upwards of $3\times$ more area than the closest autonomous approach.

Successful exploration in hazard-dense environments (Figure 4 center) necessitates the ability to identify affordance-restricting hazards, as well as the ability to plan paths that safely navigate around them. In this setting, augmenting the frontier-based approach with affordance maps increases performance by approximately 60%, which is more than 2/3 of the difference between frontier and the random baseline. Qualitatively, we observe that agents using learned affordance maps plan paths that leave a wide margin of safety around observed hazards and spend far less time stuck in areas of geometry-affordance mismatch. Through self-supervised sampling, the navigability module also learns about agent-specific locomotion capabilities, predicting when low ceilings and tall steps may restrict movement. Although RL-based exploration out-performs the frontier baseline in this scenario by learning that proximity to hazards is detrimental to reward maximization, a lack of long term planning still hinders overall exploration performance.

**Sample Efficiency.** In order to understand the effect of training set size on learned exploration, we measure exploration performance with different amounts of collected samples in the hazard-dense setting, shown in Figure 4 right. After collecting as few as 5000 training samples, the navigability module learns to recognize dynamic hazards, allowing for paths to be planned with a margin of safety. As the number of samples collected increases, exploration performance improves as well. However, as one might expect, the relative gain provided by each additional example decreases after a point. Qualitatively, we observe that 10,000 samples provides sufficient diversity to enable accurate localization of common dynamic hazards, while additional examples beyond this point help to improve detection of less commonly observed environmental hazards and accuracy near hazard boundaries. Notably, even after training on 20 times as many samples, RL-based exploration still fails to outperform our approach in this setting, illustrating a clear advantage in sample efficiency.

### 4.3 GOAL-DIRECTED NAVIGATION USING ACTIVE AFFORDANCE MAP LEARNING

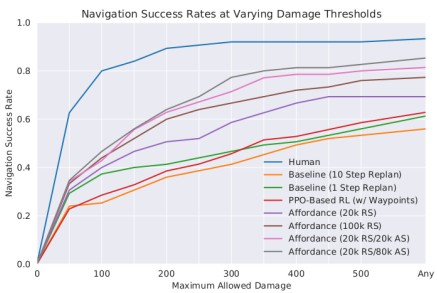

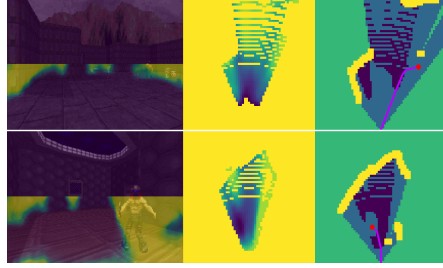

Figure 5: Comparison of navigation performance across all evaluated approaches, plotted as a function of success rate vs. maximum amount of damage permitted per trial (mean results over 5 test runs reported).

Figure 6: Examples of actively-planned trajectories that maximize label entropy along sampled locations. (**Left**) shows predicted affordances, (**Middle**) shows the projected confidence map, and (**Right**) shows the cost map used to plan the optimal path.

In order to further demonstrate the applicability and efficacy of affordance-based representations, we set up a series of 15 navigation trials, one for each map in the test set. Within each trial, the agent begins at a fixed start point and is tasked with navigating to an end goal (specified in relative coordinates) in under 1000 time steps, while minimizing the amount of damage taken along the way. Each trial is designed to be difficult, with numerous hazards and obstacles separating the start and end points, presenting a challenge even for skilled humans. Additional experimental details are discussed in Appendix A.2.

In this setting, we show that by adding semantic information obtained from affordance maps, it is possible to improve navigation performance significantly, even when employing simple geometry-based approaches that plan using A*. By introducing a navigability module trained on 100k collected samples to generate cost maps for planning, we observe a 45% improvement in overall navigation success rate, with improvements of 25% observed even when using a model trained on a dataset just one fifth the size. Even when the re-planning frequency is increased 10-fold, such that observed dynamic hazards can be treated as static obstacles more accurately, the baseline still fails to beat the affordance-augmented variant.

Additionally, we compare against results obtained by a PPO-based RL model, which is trained similarly to its counterpart discussed in Section 4.2. In order to reduce the difficulty of planning over long time horizons, we provide the model with a sequence of waypoints (extracted from the best-performing human trajectory) as an additional input, which are used as local intermediate goals that converge towards a faraway global goal. However, we observe that even with this augmented set of inputs, the RL-based approach still fails to beat any of the affordance-based methods, echoing results observed in the exploration experiments.

We also explore how active learning can be used to further improve the efficiency of self-supervised learning, by evaluating two additional models trained on samples collected from actively-planned trajectories. We show that using just 40% of the data, models employing active data collection outperform those trained using random samples alone. At the 100k total sample mark, we observe

that actively sampled models out-perform their randomly sampled counterparts by more than 10%. These results, along with comparisons to the baseline, are summarized in Figure 5; examples of actively-planned trajectories are visualized in Figure 6. Qualitatively, we observe that active trajectory sampling significantly improves temporal stability and prediction accuracy along hazard and obstacle boundaries (shown in Figure 8). These properties enable more efficient path planning, allowing the agent to move safely with tighter margins around identified hazards.

## 5 DISCUSSION

We have described a learnable approach for exploration and navigation in novel environments. Like RL-based policies, our approach learns to exploit semantic, dynamic, and even behavioural properties of the novel environment when navigating (which are difficult to capture using geometry alone). But unlike traditional RL, our approach is made sample-efficient and interpretable by way of a spatial affordance map, a novel representation that is interactively-trained so as to be useful for navigation with off-the-shelf planners. Though conceptually simple, we believe affordance maps open up further avenues for research and could help close the gap between human and autonomous exploration performance. For example, the dynamics of moving obstacles are currently captured only in an implicit fashion. A natural extension is making this explicit, either in the form of a dynamic map or navigability module that makes use of spatio-temporal cues for better affordance prediction.

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

# A APPENDIX

## A.1 EXPLORATION EXPERIMENTAL DETAILS

### A.1.1 ENVIRONMENTS

Each test map used to evaluate exploration performance is extremely large, containing sequences of irregularly shaped rooms connected by narrow hallways and openings. As it would require upwards of 10,000 time steps even for a skilled human to explore any of these environments, most of the space covered by the evaluated approaches is contained within the initial rooms next to the start point. As such, to clearly illustrate the challenges posed by semantic constraints, we choose to further categorize the test maps as either *hazard-sparse* or *hazard-dense*, based on the likelihood of encountering navigability-restricting hazards within the initial exploration area. Figure 7 shows a top-down visualization of the difference in hazard concentration between the two test subsets.

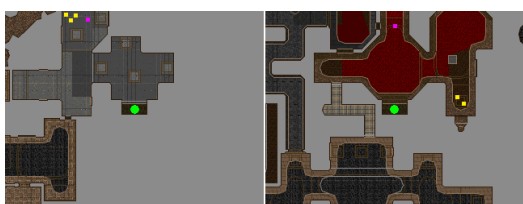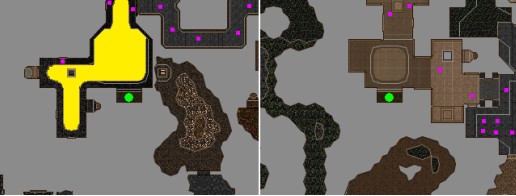

Figure 7: Top-down visualizations of initial exploration areas in hazard-sparse **(Left)** and hazard-dense **(Right)** test environments. Agent start position is marked in green, with environmental hazards marked in yellow, and initial locations of dynamic hazards marked in purple. Hazard-dense environments present a significant challenge for autonomous exploration, containing a high concentration of navigability restricting areas that must be avoided successfully.

### A.1.2 BASELINES

We next provide additional details about the baselines that we compare against:

1. *Random.* In order to show that both geometry and learning-based approaches set a competitive baseline and perform well above the bar set by random exploration, we evaluate a policy that selects between each of the available actions with uniform probability at each time step.

2. *RL-Based Exploration.* To ensure a fair comparison against RL-based exploration, we implement the approach proposed by (Chen et al., 2019) with identical hyper-parameters, maintaining a global map where each grid cell represents an $8 \times 8$ game unit area in the VizDoom world. We also introduce a new penalty term used to scale the reward by amount of damage taken in each step (set to 0.02), in order to encourage the agent to avoid hazardous regions of the map.

3. *Human.* To demonstrate that all autonomous approaches for exploration still have significant room for improvement, we asked human volunteers to explore a sequence of novel environments using a near-identical experimental setup. In order to make movement more natural, we allowed participants to execute actions for rotation and forward movement concurrently, providing a slight edge in locomotion compared to other approaches.

### A.1.3 NAVIGABILITY MODULE TRAINING

As labeled samples $\tilde{y}$ are generated by painting all pixels within a radius of a specified point in the simulated world as either navigable or non-navigable, we are more "certain" that pixels close by in world space share the same semantic constraints than those farther away. To ensure stability of training, we express this belief in the loss by weighting each pixel's contribution by its inverse Euclidean distance to the closest such point in world space.

## A.2 NAVIGATION EXPERIMENTAL DETAILS

Each trial evaluated as part of the goal-directed navigation experiments is intentionally designed to be difficult, requiring agents to navigate through areas of high hazard concentration and around

obstacles that present geometry-affordance mismatches. Indeed, none of human participants were able to complete all 15 trials successfully without taking any damage. Qualitatively, we observed that most failures in the baseline occurred when the agent attempted to path through an obstacle lower than sensor height, causing the agent to become stuck as it continually tries to path through a cell registered as free space in the geometric map. The second most common scenario that leads to degraded performance is failure to avoid dynamic hazards, causing agents to collide with monsters as they attempt to follow a nearby path to the goal.

### A.2.1 GEOMETRY-BASED NAVIGATION BASELINE

We implement the baseline approach for navigation using a modified variant of the planning and locomotion modules employed for frontier-based exploration. Global navigability maps used for planning are constructed by averaging values obtained from local maps over multiple time steps, allowing for increased temporal stability and robustness to sensor and localization noise. A simple A*-based algorithm is then employed for planning, treating the value of each cell in the global navigability map as the cost of traversing through a particular location in the environment.

In this setup, dynamic actors are treated as static obstacles within the cost map, an assumption that holds true as long as re-planning is invoked at a sufficiently high frequency. In order to evaluate the effect of re-planning frequency on navigation performance, we also evaluate a variant of the baseline approach that re-plans at $10\times$ the frequency (every step instead of every 10 steps) and observe that this results in a small improvement in navigation trial success rate, largely attributed to the reduction in latency required to respond to environmental dynamics.

### A.2.2 AFFORDANCE-AUGMENTED NAVIGATION

To evaluate the efficacy of active trajectory sampling, we evaluate affordance-augmented navigation using segmentation models trained with active data gathering. The procedure we follow is to first train a seed model using 20k random samples, before collecting an additional 20k samples actively and re-training to generate an improved version of the model. We repeat the active sample/train loop for an additional 3 iterations, building a dataset with a total size of 100k samples. Visualizations of predicted affordance maps generated by trained models after iterations 0, 2, and 4 of the sample/train loop are shown in Figure 8 and compared to a model trained using 100k randomly collected samples.

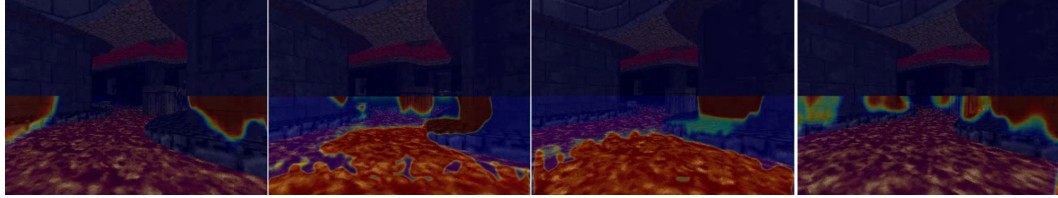

Figure 8: Comparison of affordance maps generated by models trained using datasets containing (a) 20k random samples, (b) 20k random samples + 40k active samples, (c) 20k random samples + 80k active samples, and (d) 100k random samples. Employing active learning allows models to effectively identify and localize regions containing rare environmental hazards, a feat that is difficult to achieve using random samples alone.

### A.3 CAPTURING DYNAMIC BEHAVIOR

In order to better understand how dynamic behavior is captured using our affordance-labeling approach, we pose a scenario where a dynamic actor moves from point A to point B, and collides with the agent at point B. In this scenario, all observations of point B (including those collected pre-collision) will be labelled as hazardous, potentially mapping to an image region near the dynamic actor rather than the actor itself. We will next describe one approach for explicitly modeling such moving obstacles, and then justify why our current approach implicitly captures such dynamics.

**Explicit Approach.** In principle, our self-supervised labeling system can be modified to replace naive back-projection with an explicit image-based tracker (keeping all other components fixed). Essentially, labeled patches can be tracked backwards from the final timestep at which they are

identified as hazardous (since those prior visual observations are available at sample time) to obtain their precise image coordinates when backprojecting to prior timesteps.

**Implicit Approach.** Even without image-based tracking, our pipeline implicitly learns to generate larger safety margins for visual signatures that have been associated with dynamic behavior. Essentially, our system learns to avoid regions that are spatially close to dynamic actors (as seen in Figure 9). Such notions of semantic-specific safety margins (e.g., autonomous systems should use larger safety margins for pedestrians vs. roadside trash cans) are typically hand-coded in current systems, but these emerge naturally from our learning-based approach. As we found success with an implicit encoding of dynamics, we did not experiment with explicit encodings, but this would certainly make for interesting future work.

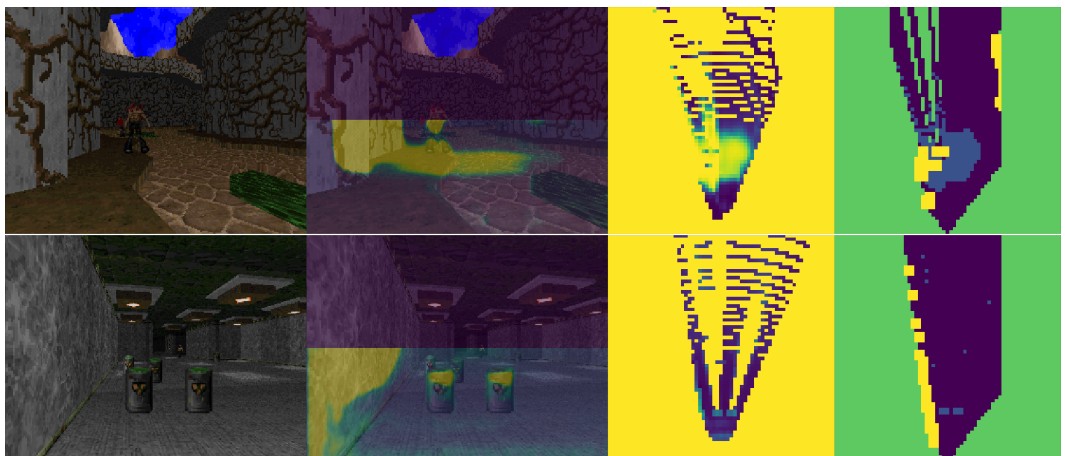

Figure 9: Examples of learned margins for visual signatures associated with dynamic actors. From left to right: the first image shows a RGB view of the scene, the second image shows predicted affordances $\hat{y}$ overlaid on top of the RGB view, the third image shows the projected confidence map, and the last image shows the cost map used to plan the optimal path. From the first example, it can be seen that regions that are spatially close to dynamic actors are associated with higher traversal costs in the final cost map, akin to a "margin of safety". The second example shows that static hazards/obstacles such as barrels are not associated with substantial affordance margins.

## A.4 REAL-WORLD APPLICABILITY

During the sampling stage of our proposed method, we employ a "trial and error"-style approach (similar to RL-based methods) that could lead to potentially hazardous situations in real-world robotics settings if deployed in certain configurations. However, we argue that this is not an unreasonable way of collecting data and that there exist both common and practical solutions for risk mitigation that have already been widely deployed in the real-world.

Framing the current state of self-driving research within the context of our work, we can view all autonomous vehicles today as being within the sampling stage of a long-term active learning loop that ultimately aims to enable L4 autonomy. Almost every one of these vehicles on public roads today is equipped with one, if not multiple safety operators who are responsible for disengaging autonomy and taking over when the system fails to operate within defined bounds for safety and comfort. Moreover, each of these takeover scenarios is logged and used to improve the underlying models in future iterations of this learning loop (Dixit et al., 2016). Indeed, in this scenario, the safety operator serves the purpose of the feedback sensor and can ultimately be removed at test time, once the autonomous driving model has been deemed safe.

In less safety critical scenarios, such as closed course or small-scale testing, the role of the safety driver could be replaced with some form of high-frequency, high-resolution sensing such as multiple short-range LIDARs. These feedback sensors can be used to help the robot avoid collisions during the initial stages of active training, stopping the agent and providing labels whenever an undesirable state is entered. Importantly, since data from these expensive sensors is not directly used as an input

by the model, they can be removed once a satisfactory model has been trained; production-spec robots are free to employ low-cost sensing without the need for high-cost feedback sensors.

Additionally, there exist many scenarios in which feedback sensors can help label examples without the need to experience catastrophic failures such as high-speed collisions. One example is the discrepancy between wheel speed sensor values, which can be used to detect loss of traction on a wheeled robot when travelling over rough or slippery surfaces. By collecting observation-label pairs, we could then learn affordance maps to help such a robot navigate over the smoothest terrain.

Finally, we would like to emphasize that in scenarios where it is difficult to obtain oracle-labelled data and "trial and error" approaches are employed by necessity, we have shown that our proposed approach is many times more sample efficient that previous PPO-based reinforcement learning approaches for mobile robotics (Chen et al., 2019) (which also suffer from the same types of problems). If collecting a sample is costly due to the burden of operational hazards, we argue that a reduction in the number of samples required translates to an improvement in overall safety.

## A.5 CODE

As the proposed system contains a large number of moving parts and tunable hyper-parameters, we plan to release modular open-source code for the described system at the following location: https://github.com/wqi/A2L.

## A.6 ADDITIONAL VISUALIZATIONS

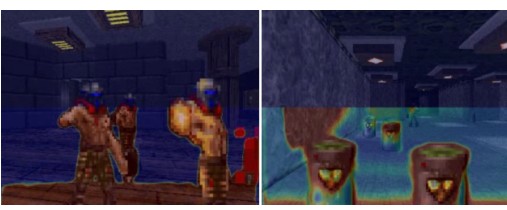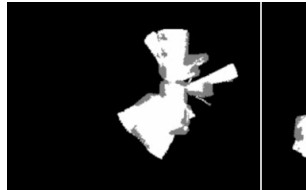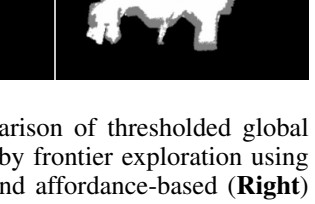

Figure 10: Examples of affordance maps $\hat{y}$ predicted by the navigability module, showing accurate localization of semantic constraints within the scene. (**Left**) contains dynamic hazards in the form of monsters and (**Right**) contains areas of geometry-affordance mismatch, in the form of barrels shorter than sensor height.

Figure 11: Comparison of thresholded global maps constructed by frontier exploration using geometry (**Left**) and affordance-based (**Right**) representations in the same environment. In this setting, semantic representations help the agent take less damage over time, allowing for more area to be explored during the episode.

