# OpenReview forum: "Learning to Move with Affordance Maps"
_ICLR.cc/2020/Conference — Accept (Poster)_

### Official Review · AnonReviewer1 · 2019-10-23
**Official Blind Review #1**

**Rating:** 6

**Review:**

The paper proposes to learn affordance maps: a method to judge whether a certain location is accessible. This is done by distilling a series of "trial and error" runs and the relation of a pixel in the image/depth plane to a corrdinate into a model.

I like the idea and think the paper should be accepted. The idea to use trial and error (something I prefer to self-supervision, which is used differently in many contexts, I believe) to obtain a data set for learning a model is nice and very practical.

Some concerns that I think should be adressed.

- The term information gain is used wrongly. The entropy of class labels is not infogain. Infogain is the expected KL of the model posterior from the model prior. Please correct this. See [1, 2].
- *Learning* a model of the environment and using it for navigation/exploratin has also been tackled recently by [1]. I think the authors should draw connetions to that work.
- Self-supervision has recently been proposed by Lecun as a subsitute (of sorts) for unsupervised learning. What he means is that a part of the data is used to predict another part of the data. I have no hard feelings about the term, personally preferring unsupervised, but the authors should be aware of the name clash.

I wonder how the authors envision to extend this method to real scenarios. The "trial and error" method is clearly not viable for robotics setups, as hazards are costly. It would be nice if the authors could give there perspective on things.

[1] Depeweg et al, "Decomposition of Uncertainty in Bayesian Deep Learning for Efficient and Risk-sensitive Learning", Proceedings of the 35th International Conference on Machine Learning
[2] Mirchev et al, "Approximate Bayesian Inference in Spatial Environments" in proceedings of Robotics: Science and Systems XV.

**Experience Assessment:**

I have published one or two papers in this area.

**Review Assessment: Checking Correctness Of Derivations And Theory:**

I assessed the sensibility of the derivations and theory.

**Review Assessment: Checking Correctness Of Experiments:**

I assessed the sensibility of the experiments.

**Review Assessment: Thoroughness In Paper Reading:**

I read the paper at least twice and used my best judgement in assessing the paper.

---

> ### Author Response · Authors · 2019-11-08
> **Response to Reviewer 1 (1/2)**
>
> Dear Reviewer 1,
>
> Thank you very much for your positive and constructive feedback, we are glad to hear that you found our work to be practical and interesting. To address your comments and concerns:
>
> [Terminology]
> We have updated our paper to correct our use of the term “information gain”, apologies for any confusion with our choice of terminology in the original text. Additionally, thank you for bringing the multiple definitions associated with the term “self-supervision” to our attention, we will be sure to keep this in mind going forward.
>
> [Related Work]
> >> *Learning* a model of the environment and using it for navigation/exploration has also been tackled recently by [2]. I think the authors should draw connections to that work.
>
> Mirchev et al. propose an interesting method of learning a generalized spatial representation that can be used for both navigation and exploration. The approach employs a deep sequential generative model and roughly metric map to reconstruct observations using pose-based attention, sharing our view that structured intermediate representations are important. Thank you for pointing out this piece of related work, we have cited and briefly compared our approach in the updated version of the paper.
>
> We believe that the primary similarities between our work and that of [2] are that both approaches construct metric maps that contain information used to infer affordances at particular locations in world space, using models trained with dense supervision. However, our approach employs a predictive model, rather than an attention-based generative model. Additionally, we plan directly on top of a metric cost map, whereas the map employed by [2] is in latent space, with planning for navigation occurring in belief space. Another difference is that [2] employs observation reconstruction as a training signal, whereas we employ sensor feedback coupled with back-projection. Finally, a major difference is that the concept of affordance in our evaluation environments depends heavily on both dynamics and semantics, two types of constraints that [2] does not address (as affordances in their evaluation are defined solely by geometry).

---

> ### Author Response · Authors · 2019-11-08
> **Response to Reviewer 1 (2/2)**
>
> [Real World Robotics Deployment]
> >> The "trial and error" method is clearly not viable for robotics setups, as hazards are costly. It would be nice if the authors could give their perspective on things.
>
> To share our perspective, while it is true that taking a “trial and error” approach to sampling could lead to potentially hazardous situations in real-world robotics settings, we believe that this is not an unreasonable way of collecting data and that there exist practical solutions for risk mitigation that have already been widely deployed.
>
> Framing the current state of self-driving research within the context of our work, we can view all autonomous vehicles today as being within the “sampling” stage of a long-term active learning loop that ultimately aims to enable L4 autonomy. Almost every one of these vehicles on public roads today is equipped with one, if not multiple safety operators who are responsible for disengaging autonomy and “taking over” when the system fails to operate within defined bounds for safety and comfort. Moreover, each of these “takeover” scenarios is logged and used to improve the underlying models in future iterations of this learning loop [3]. Indeed, in this scenario, the safety operator serves the purpose of the “feedback sensor” and can ultimately be removed at “test time”, once the autonomous driving model has been deemed safe.
>
> In less safety critical scenarios, such as closed course or small-scale testing, the role of the safety driver could be replaced with some form of high-frequency, high-resolution sensing such as multiple short-range LIDARs. These feedback sensors can be used to help the robot avoid collisions during the initial stages of active training, stopping the agent and providing labels whenever an undesirable state is entered. Importantly, since data from these expensive sensors is not directly used as an input by the model, they can be removed once a satisfactory model has been trained; production-spec robots are free to employ low-cost sensing without the need for high-cost feedback sensors.
>
> Additionally, there exist many scenarios in which feedback sensors can help label examples without the need to experience catastrophic failures such as high-speed collisions. One example is the discrepancy between wheel speed sensor values, which can be used to detect loss of traction on a wheeled robot when travelling over rough or slippery surfaces. These labels could them be used to train a model that generates learned affordance maps to help such a robot navigate over the smoothest terrain.
>
> Finally, we would like to emphasize that in scenarios where it is difficult to obtain oracle-labelled data and “trial and error” approaches are employed by necessity, we have shown that our proposed approach is many times more sample efficient that previous PPO-based reinforcement learning approaches for mobile robotics [4] (which, as R3 notes, also suffer from the same types of problems). If collecting a sample is costly due to the burden of operational hazards, we believe that a reduction in the number of samples required translates to an improvement in overall safety.
>
> [3] Dixit, V. V., Chand, S., & Nair, D. J. (2016). Autonomous vehicles: disengagements, accidents and reaction times. PLoS one, 11(12), e0168054.
> [4] Chen, T., Gupta, S., & Gupta, A. (2019). Learning exploration policies for navigation. ICLR 2019

---

> > ### Comment · AnonReviewer1 · 2019-11-15
> > **Thanks.**
> >
> > Thanks for the clarification.

---

### Official Review · AnonReviewer2 · 2019-10-23
**Official Blind Review #2**

**Rating:** 6

**Review:**

This paper presents an approach for navigating and exploring in environments with dynamic and environmental hazards that combines geometric and semantic affordance information in a map used for path planning.

Overall this paper is fairly well written.  Results in a VizDoom testbed show favorable performance compared to both frontier and RL baselines, and the author's approach is more sample-efﬁcient and generalizable than RL-based approaches.

I wouldn't consider any particular aspect of this paper to be that novel, but it is a nice combination of leveraging active self-supervised learning to generate spatial affordance information for fusion with a geometric planner.

As humans show the best performance on the tasks, it might be worth considering learning a policy from human demonstrations through an imitation learning approach.

**Experience Assessment:**

I have read many papers in this area.

**Review Assessment: Checking Correctness Of Derivations And Theory:**

N/A

**Review Assessment: Checking Correctness Of Experiments:**

I assessed the sensibility of the experiments.

**Review Assessment: Thoroughness In Paper Reading:**

I read the paper at least twice and used my best judgement in assessing the paper.

---

> ### Author Response · Authors · 2019-11-08
> **Response to Reviewer 2**
>
> Dear Reviewer 2,
>
> Thanks for your constructive feedback! We agree that imitation learning from human demonstration is interesting, and provide some thoughts on how it could fit with our framework.
>
> Our experimental results in the paper demonstrate that humans are exceptionally good at both exploration and navigation, beating all autonomous approaches even without any previous experience performing the task at hand. We hypothesize that this performance is largely explained by strong priors that our human participants have built up over time from previous video games; e.g., recognition that “red lava is probably hazardous” speeds up learning.
>
> The most straightforward way to incorporate imitation learning would be to train a model that maps directly from visual inputs to action outputs, attempting to mimic human actions from a training distribution. However, this approach is associated with a well-known drawback of imitation learning: once the model makes a mistake and veers off policy, it is difficult to recover. For example, once the agent gets stuck in a corner, it may not have encountered a training sample that reveals how to escape (because humans are unlikely to make such mistakes).
>
> Instead, we point out that imitation learning can be applied within our factored approach for learning affordance maps. Specifically, we can make use of expert strategies for exploring and actively sampling new environments (for which previously-acquired priors are of little help). Dubey et al. [5] ingeniously create a 2D-platformer gaming world where visual signatures are systematically masked to eliminate the applicability of visual priors, making it dramatically harder for humans to navigate. In such environments, humans must “probe” each new texture in order to understand its effects, analogous to the sampling process employed by our agent in the active learning loop. Given that our approach requires the collection of samples numbering in the thousands and humans have been shown to adapt to novel environments with far fewer, it seems promising to apply imitation learning for the task of learning of better sampling policies.
>
> [5] Dubey, R., Agrawal, P., Pathak, D., Griffiths, T. L., & Efros, A. A. (2018). Investigating human priors for playing video games. arXiv preprint arXiv:1802.10217.

---

### Official Review · AnonReviewer3 · 2019-10-24
**Official Blind Review #3**

**Rating:** 6

**Review:**

The paper proposes an interesting, and to the best of my knowledge novel, pipeline for learning a semantic map of the environment with respect to navigability, and simultaneously uses it for further exploring the environment.

The pipeline can be summarized as follows: Navigate somewhere using some heuristic. When navigation "works", as well as when encountering something "negative", back-project that into past frames, and label the corresponding pixels as such: either positive or negative. This generates a collection of partially densely labelled images, on which a segmentation network can be learned that learns which part of the RGBD input are navigable and which should be avoided. For navigation, navigability of the current frame is predicted, and that prediction is down-projected into an "affordance map" that is used for navigation. One experiment confirms the usefulness of such an affordance map.


I am marking weak reject currently because of the following concerns, which might be me just missing something. On the one hand, I am glad to see something that is not just blind "end to end RL with exploration bonus", sounds reasonable, and works well. On the other hand, I do have several major concerns about the method, outlined as follows:

1. How can this approach work for moving obstacles? Let's say a monster walks from point A to point B, and collides with the agent at point B. Then, point B is marked as a hazard, but in the previous frames, the monster is not located at point B, and thus an image region that does not contain the monster is marked as hazard. Am I missing something here?
2. The method does not seem practical for actual mobile robots, only for in-game or in-simulation agents. The reason being that in order to learn "robot should not bump into baby", the robot actually needs to bump into multiple babies in order to collect data about that hazard. To be fair, blind "PPO+exploration bonus" suffers from the same problem, but in this paper, the whole motivation is about mobile robots (at least that was my impression after reading it).

Furthermore, I do not think I would be able to reproduce any of the experiments, as many details are missing. Will code be released?


###### Post-rebuttal update

I am happy with the author's response to my concerns, and they have included corresponding discussions in their paper. Thus, I am improving my rating to recommend acceptance of this paper to ICRL2020.

**Experience Assessment:**

I have read many papers in this area.

**Review Assessment: Checking Correctness Of Derivations And Theory:**

N/A

**Review Assessment: Checking Correctness Of Experiments:**

I carefully checked the experiments.

**Review Assessment: Thoroughness In Paper Reading:**

I read the paper at least twice and used my best judgement in assessing the paper.

---

> ### Author Response · Authors · 2019-11-08
> **Response to Reviewer 3**
>
> Dear Reviewer 3,
>
> Thank you for the constructive and detailed feedback, we are happy to address your questions and concerns, and will update the paper to improve the clarity of exposition.
>
> [Dealing with Environmental Dynamics]
> >> How can this approach work for moving obstacles? Let's say a monster walks from point A to point B, and collides with the agent at point B. Then, point B is marked as a hazard, but in the previous frames, the monster is not located at point B, and thus an image region that does not contain the monster is marked as hazard. Am I missing something here?
>
> This is a great question! In the scenario that you have posed, it’s true that if the monster moves between when the agent was at point A and when the agent reached point B, the hazard label will map to an image region near the monster, rather than the monster itself. Let us first describe one approach for explicitly modeling such moving obstacles, and then justify why our current approach implicitly captures such dynamics.
>
> Explicit approach: in principle, one can modify our self-supervised labeling system to replace naive back-projection with an explicit image-based tracker (keeping all other components fixed). Essentially, track hazards backward from the final timestep at which they are identified as hazardous (since those prior visual observations are available at sample time) and obtain their precise image coordinates when backprojecting those prior timesteps.
>
> Implicit approach: Even without image-based tracking, our pipeline implicitly learns to associate larger safety margins for visual signatures that are dynamic. Essentially, our system learns to avoid regions that are spatially nearby dynamic objects (please refer to Fig 11 in the updated appendix). Such notions of semantic-specific safety margins (e.g., autonomous systems should use larger safety margins for people vs roadside trash cans) are typically hand-coded in current systems, but these emerge naturally from our learning-based approach.
>
> Because we found success with an implicit encoding of dynamics, we did not experiment with explicit encodings. But we agree that it would be interesting future work.
>
> [Real World Generalization]
> >> The method does not seem practical for actual mobile robots, only for in-game or in-simulation agents. The reason being that in order to learn "robot should not bump into baby", the robot actually needs to bump into multiple babies in order to collect data about that hazard. To be fair, blind "PPO+exploration bonus" suffers from the same problem, but in this paper, the whole motivation is about mobile robots (at least that was my impression after reading it).
>
> This is another great question! We refer R3 to our second response to R1, who shared a very similar concern.
>
> [Code Release]
> >> Will code be released?
> We understand that the described system contains a rather large number of moving parts and hyper-parameters, which could be challenging to reproduce. To address this concern, we plan to release code for the full system with modular components for sampling, model training, map construction, planning, and locomotion. We hope that our modular code will enable researchers to re-use modules and swap out individual components to try out new approaches.

---

> > ### Comment · AnonReviewer3 · 2019-11-15
> > **Thank you for addressing each of my points.**
> >
> > - Your answer regarding dynamics makes a lot of sense. If you could include some textual discussion similar to your answer into the paper, I believe it would help a lot in contextualizing it. It's fine if that happens in the Appendix, as long as you refer to it from the main paper (maybe in the intro).
> >
> > - I am also happy with your answer regarding real-world applicability (in your answer to R1), and again I highly recommend adding such discussion into the main paper.
> >
> > - Finally, it would be good to mention open-sourcing in the paper itself.
> >
> > If you are able to include these in the paper, I am happy to update my recommendation to accepting this paper.

---

> > > ### Author Response · Authors · 2019-11-15
> > > **Will make all requested changes to address discussed topics.**
> > >
> > > Thank you again for the insightful feedback!
> > >
> > > We are happy to make all of the requested changes and will update the next revision of the paper with additional discussion about handling of dynamics and real-world applicability, along with plans to release open-source code.
> > >
> > > As the discussion period is ending very soon, we will initially include such discussion in the appendix and will move to integrate these topics into the main text afterwards.

---

> > > ### Author Response · Authors · 2019-11-15
> > > **Paper updated to address discussed topics.**
> > >
> > > Hi Reviewer 3,
> > >
> > > As requested, we've updated the paper to include additional discussion about handling of dynamics, real-world applicability, along with plans to release open source code in the appendix.
> > >
> > > We've also updated the main text to refer to the appendix for additional details where relevant.

---

### Decision · Program_Chairs · 2019-12-19

**Decision:**

Accept (Poster)

**Comment:**

This paper presents a framework for navigation that leverages learning spatial affordance maps (ie what parts of a scene are navigable) via a self-supervision approach in order to deal with environments with dynamics and hazards. They evaluate on procedurally generated VizDoom levels and find improvements over frontier and RL baseline agents.

Reviewers all agreed on the quality of the paper and strength of the results. Authors were highly responsive to constructive criticism and the engagement/discussion appears to have improved the paper overall. After seeing the rebuttal and revisions, I believe this paper will be a useful contribution to the field and I’m happy to recommend accept.